# Dissection of the Genetic Basis of Resistance to Stem Rot in Cultivated Peanuts (*Arachis hypogaea* L.) through Genome-Wide Association Study

**DOI:** 10.3390/genes14071447

**Published:** 2023-07-14

**Authors:** Liying Yan, Wanduo Song, Zhihui Wang, Dongyang Yu, Hari Sudini, Yanping Kang, Yong Lei, Dongxin Huai, Yuning Chen, Xin Wang, Qianqian Wang, Boshou Liao

**Affiliations:** 1Key Laboratory of Oil Crops Biology and Genetic Improvement, Ministry of Agricultural and Rural Affairs, Oil Crops Research Institute, Chinese Academy of Agricultural Sciences, Wuhan 430062, China; yanliying@caas.cn (L.Y.); songwanduo@caas.cn (W.S.); wangzhihui0229@126.com (Z.W.); yudongyang0730@163.com (D.Y.); kangyanping@caas.cn (Y.K.); leiyong@caas.cn (Y.L.); dxhuai@caas.cn (D.H.); ynchen@126.com (Y.C.); wangxin456_2000@163.com (X.W.); wqqwaityou@163.com (Q.W.); 2International Crops Research Institute for the Semi-Arid Tropics (ICRISAT), Patancheru 502324, India; h.sudini@cgiar.org

**Keywords:** peanut, stem rot, resistance, genome-wide association study (GWAS), significant SNP

## Abstract

Peanut (*Arachis hypogaea*) is an important oilseed and cash crop worldwide, contributing an important source of edible oil and protein for human nutrition. However, the incidence of stem rot disease caused by *Athelia rolfsii* poses a major challenge to peanut cultivation, resulting in significant yield losses. In this study, a panel of 202 peanut accessions was evaluated for their resistance to stem rot by inoculating plants in the field with *A. rolfsii*-infested oat grains in three environments. The mean disease index value of each environment for accessions in subsp. *fasitigiate* and subsp. *hypogaea* showed no significant difference. Accessions from southern China displayed the lowest disease index value compared to those from other ecological regions. We used whole-genome resequencing to analyze the genotypes of the accessions and to identify significant SNPs associated with stem rot resistance through genome-wide association study (GWAS). A total of 121 significant SNPs associated with stem rot resistance in peanut were identified, with phenotypic variation explained (PVE) ranging from 12.23% to 15.51%. A total of 27 candidate genes within 100 kb upstream and downstream of 23 significant SNPs were annotated, which have functions related to recognition, signal transduction, and defense response. These significant SNPs and candidate genes provide valuable information for further validation and molecular breeding to improve stem rot resistance in peanut.

## 1. Introduction

Peanut (*Arachis hypogaea* L.) is a significant source of edible oil and protein and was planted on approximately 30 million hectares with a total production of around 50 million tons worldwide in 2022 (https://www.fao.org/faostat, accessed on 8 March 2023). China is one of the leading peanut-producing countries, with a planting area of 4.74 million hectares and the largest peanut production of approximately 18 million tons in 2022 (http://www.stats.gov.cn, accessed on 8 March 2023). Stem rot, also known as southern blight, southern stem rot, or white mold, is caused by *Athelia rolfsi* (Curzi) C.C. Tu & Kimbr. (*Sclerotium rolfsii* Sacc.), has been a constraint on peanut production in China in the past twenty years, and has become one of the most important soil-borne diseases of peanut in many producing areas, particularly in the Liaoning, Jiangxi, Guangdong, and Henan provinces [1,2,3,4]. Yield losses due to stem rot in peanuts range from 10% to 30% and can reach up to 80% in heavily infested fields [5,6]. Although cultural methods, fungicides, and biological agents may reduce the risk of infection by stem rot disease, they do not provide sufficient protection, and they require additional labor and financial input. The use of stem-rot-resistant cultivar is an effective and sustainable approach to manage the disease [7,8,9], while traditional methods of breeding stem-rot-resistant peanut varieties have low efficiency and are time-consuming. Utilization of molecular markers linked with stem rot disease resistance might accelerate the process of developing resistant varieties.

Stem rot resistance in peanut is a complex trait that is controlled by multiple genes and influenced by environments [10]. Previous studies have attempted to dissect the genetic basis of resistance to stem rot in peanut, and some quantitative trait loci (QTLs) linked to stem rot resistance have been identified. The first QTL linked to stem rot resistance in peanut was identified using an F2 population derived from TG37A (Susceptible) X NRCG CS85 (Resistant) with SSR markers [11]. Later on, Dodia et al. (2019) identified seven major QTLs in the F6 generation of the same recombinant inbred population (RIL) derived from TG-37Aand NRCG CS85, with variation explained ranging from 5.5% to 8.5% [10]. Using another RIL population derived from a cross of NC3033 (resistant) and Tifrunner (susceptible), Luo et al. (2020) identified a total of 33 additive QTLs for stem rot resistance using SNP and SSR markers [12], and Cui et al. (2020) identified another two QTLs linked to stem rot resistance using the QTL-seq method [13]. The QTLs linked to stem rot resistance were identified using bi-parental populations with a narrow genetic background. To discover new genes or QTLs associated with stem rot resistance in peanut, more resistant sources to stem rot resistance should be used. Association analysis employs natural populations to discover genomic regions associated with target traits in a relatively high-resolution and unbiased manner in broad-based and diverse accessions [14].

GWAS, which stands for whole-genome association study, is an observational study that investigates whether any genetic variants in a genome-wide set of variants is associated with specific traits in different individuals [15]. This approach has proven to be a powerful tool for detecting markers that are closely linked to QTLs based on the principle of linkage disequilibrium between genetic markers and QTLs [16]. Recent advances in genomic sequencing technologies, coupled with the availability of tetraploid genome sequences in *Arachis* species [17,18,19], have enabled high-throughput genotype data to be combined with phenotypic data for peanut breeding and genetics research. Such advances have also enabled the discovery of marker–trait associations through GWAS. In recent years, GWAS have been successfully conducted in peanut to unravel the genetic basis of some traits, such as oil content [20], domestication [21], plant characteristics [14,22,23], yield-related traits [24,25], and disease resistance [23,26,27,28]. Although stem rot is a destructive disease in peanut, no studies have yet been conducted using GWAS to identify QTLs related to stem rot resistance. 

This study utilized whole-genome resequencing of 202 peanut accessions, mainly from Chinese varieties, to mine high-quality SNPs distributed throughout the peanut genome. GWAS was performed to identify genomic regions associated with resistance against stem rot disease and to discover potential candidate genes within the QTLs regions in peanut. The SNP markers and candidate genes related to stem rot resistance identified in this study have the potential to assist peanut breeding in developing resistance varieties against this disease.

## 2. Materials and Methods

### 2.1. Plant Materials and Field Evaluation of Stem Rot Resistance

A total of 202 peanut accessions were included in the current study, of which 164 cultivars were collected from 18 provinces of China, and 38 cultivars were collected from other countries. There are 110 accessions belonging to subspecies *fastigiata* and 92 accessions belonging to subspecies *hypogaea* (Appendix A). The 202 accessions were planted on 6 May 2019 and on 10 May 2021 at the Wuchang Experiment Station (WES) and on 10 May 2021 at the Yangluo Experiment Station (YES) of the Oil Crops Research Institute using a randomized complete block design with three replicates. Each plot was set with a 2.5 m row length and 0.33 m row space, with a seedling rate of 15 seeds per row. Field inoculation was performed using one moderately virulent strain (WH1) of *A. rolfsii*, as described by Fan et al., 2020 [29]. The inoculum, *A. rolfsii*-infested oat grains as mentioned by Fan et al. (2020) [29] and Shokes et al. (1998) [30], was prepared by placing sterile sclerotia of *A. rolfsii* WH1 on a potato dextrose agar (PDA) medium and incubating for 48 h at 30 °C. Five mycelium discs were transferred to 300 g autoclaved oat grains in a 1 L flask. The culture was incubated at 30 °C under dark for 4~5 days and shaken daily until the oat grains were completely covered by mycelium. The oat grain culture was placed in a clean kraft envelope, dried in an oven at 40 °C for a week, and kept under 4 °C for further use. Approximately 12 healthy plants per plot were inoculated with mycelium-infested oat grains 80–90 days after sowing. Around one gram of oat grains culture was spread on the soil surface around the stem base of each plant. Field soil was irrigated to water capacity after inoculation for 2–3 consecutive days as mentioned by Yan et al., 2022 [31]. Disease rating was conducted around 14 to 21 days after inoculation (DAI). Each plant was rated individually on a visual scale of 0 to 4 for the severity of wilting, as described by Shokes et al. (1998) [30], where 0 = no symptoms, 1 = symptom only on the stems, 2 = less than 25% wilting foliage, 3 = 25 to 50% wilting foliage, and 4 = more than 50% wilting foliage. The disease index (DI) was calculated according to Yan et al. (2022) [31] as follows: DI = [(1 × number of plants classified in scale 1) + (2 × number of plants classified in scale 2) + (3 × number of plants classified in scale 3) + (4 × number of plants classified in scale 4)]/(4 × total number of plants) × 100. The mean disease index of three replicates in each environment was used for phenotypic data for GWAS analysis. The phenotypic data collected in WES were denoted as A followed by the year, i.e., 2019-A-DI and 2021-A-DI, and those in YES were denoted as B followed by the year, i.e., 2021-B-DI. Therefore, in total, three sets of phenotypic data were collected. The phenotypic distribution was plotted using OriginLab Origin 2019b (OriginLab Corporation, Northampton, MA, USA). All statistical analyses were conducted using SPSS 25 (IBM SPSS Statistics, Chicago, IL, USA). 

### 2.2. Samples Preparation and Genotyping

Young leaves from the 202 peanut accessions were collected from field-grown plants for DNA extraction. Genomic DNA was extracted using a Hi-DNA secure Pant Kit (Tiangen, Beijing, China). The quality and quantity of DNA were assessed using a NanoDrop 2000 (Thermos Fisher Scientific Inc., Wilmington, DE, USA) and 1% agarose gel electrophoresis. Whole-genome resequencing (WGS) of the panel was performed using the BGISEQ-500/MGISEQ-2000 platform. Clean data were obtained by removing adaptors and low-quality reads with Soapnuke software (BGI Company, Hong Kong, China). High-quality unique reads were then mapped onto the reference genome of tetraploid cultivated peanut (*A. hypogaea*) Tifrunner [17] using the Burrows–Wheeler Aligner (BWA) [32] with the following parameters: —t 8—k 19—M—R. Alignment duplications were removed using SAMTools [33] and Picard Tooklit (https://broadinstitute.github.io/picard/sss, accessed on 20 October 2022). SNP identification was carried out using GATK [34] with the following parameters: MQ > 30, QD < 1.5, —max missing 0.9—min alleles 2—max alleles 2—maf 0.05, and -cluster-window size 10. A total of 3,034,414 SNPs markers were retained after filtering out SNPs with genotyping error, a call rate <0.90 or minor allele frequency <0.05.

### 2.3. Population Structure Analysis and Phylogenetic Tree Construction

High-quality SNPs markers obtained from the DNA samples of the 202 peanut accessions were used to conduct principal component analysis (PCA), population structure analysis, phylogenetic trees construction, and relative kinship analysis. PCA was performed using EIGENSOFT software (https://mybiosoftware.com/eigensoft-population-structure-eigenanalysis-stratification.html, accessed on 20 October 2022). The population structure of the peanut panel was assessed using fastStructure software [35] with K-values ranging from 0 to 7. Pairwise distance among the 202 accessions were calculated using the *p*-distance model, and a phylogenetic tree with 1000 bootstraps was constructed using the maximum likelihood methods in MEGA6 [36].

### 2.4. Genome-Wide Association Analysis and Candidate Genes’ Predication

SNPs with a minor allele frequency (MAF) higher than 0.01 were employed for linkage disequilibrium analysis. LD blocks were visualized using Haploview 4.2 [37] following the protocol of Sardos et al. [38]. The genome-wide association analysis was performed with high-integrity SNPs and phenotypic data from three environments of 202 peanut accessions using rMVP software [39] with three models, namely the generalized linear model (GLM), which uses population structure as a covariate [40]; the mixed linear model (MLM), which incorporates both the population structure and kinship among the individuals to adjust association tests on markers [41]; and fixed and random model circulating probability unification (FarmCPU), which performs marker tests with associated markers as covariates in a fixed-effect model and performs optimization on the associated covariate markers in a random-effect model separately [42]. A significant threshold of *p* < 0.01 was used to determine the association. Candidate genes located within a 100 kb region upstream or downstream of the significant SNPs were identified from the *A. hypogaea* cv. Tifrunner: assembly and annotation (gnm1.ann1) available at https://peanutbase.org (accessed on 20 October 2022). The *r*^2^ value was used to explain the phenotypic variation of each marker following the methods described by Zhang et al. (2017) [21].

## 3. Results

### 3.1. Phenotypic Variation among Peanut Accessions

The disease-resistance parameters showed a wide range of phenotypic variation both within and across three environments, and they exhibited a nearly normal distribution (Figure 1; Table 1). In the WES of 2019 (2019-A), the disease index ranged from 27.03 to 90.10 with an average of 49.68. In the WES of 2021 (2021-A), the disease index ranged from 22.97 to 91.54 with an average of 42.77, while in the YES of 2021 (2021-B), the disease index ranged from 34.26 to 94.29 with an average of 59.92. The coefficient of variation (CV) for the disease index values of 2019-A, 2021-A, and 2021-B were 24.62, 25.92, and 20.47, respectively. The disease index in 2021-A showed significant correlation with that of 2019-A (*r* = 0.39, *p* < 0.01) and with that of 2021-B (*r* = 0.41, *p* < 0.01), while it did not show significant correlation with that of 2021-B (*r* = 0.24) (Appendix A). ANOVA analysis of the 202 accessions indicated that the disease index was significant (*p* < 0.01) for environment, genotype, and genotype × environment interaction (Appendix A). The broad-sense heritability estimated for the disease index was 0.67 in a combined ANOVA across the three environments (Table 1), indicating that resistance was heritable.

Peanut accessions were classified into two subspecies, *fastigiata* and *hypogaea*. In this study, 110 peanut accessions belonged to *fastigiata* subspecies, and 92 accessions belonged to *hypogaea* subspecies (Appendix A). The mean disease index of accessions in subsp. *fastigiata* (52.80) was lower compared to accessions in subsp. *hypogaea* (54.89) across three environments, and the disease index of accessions in subsp. *fastigiata* (47.86 in 2019-A, 52.06 in 2021-A, 58.47 in 2021-B) was also lower compared to accessions in subsp. *hypogaea* (49.34 in 2019-A, 53.31 in 2021-A, 61.95 in 2021-B) in each environment, although the difference was not significant (Table 2). The peanut panel used in the current study was divided into four groups based on the geographical location of each accession: southern China (Guangdong, Guangxi, Fujian and Hainan); the Yangtze River region (Yunnan, Guizhou, Sichuan, Hubei, Hunan, Jiangxi, Anhui, Jiangxi, and Jiangsu); northern China (Liaoning, Hebei, Shandong, Shanxi, Henan); and others (ICRISAT, America, Africa) (Appendix A). The mean disease index of accessions in southern China (48.89) was the lowest across three environments and significantly differed from accessions from the Yangtze River region and other countries, but it did not significantly differ from accessions from northern China. The disease index of accessions in southern China was also the lowest in each environment and was significantly different from accessions from the Yangtze River region and other countries in 2019-A, 2021-A, and 2021-B, as well as from accessions from northern China in 2021-B; however, it did not significantly differ from accessions from northern China in 2019-A and 2021-A (Table 3). Moderately resistant and highly susceptible peanut accessions among the 202 accessions were identified based on the mean disease index across three environments (Appendix A). Seven accessions with a mean disease index lower than 40 were considered moderately resistant, including CE036, CE064, CE072, CE084, CE103, CE104, and CE142. Four accessions, CE032, CE133, CE133, and CE171, with a mean disease index higher than 80 were regarded as highly susceptible. The moderately resistant accessions included four accessions belonging to *fastigiata* subspecies and three belonging to *hypogaea* subspecies. The four highly susceptible accessions all belonged to *hypogaea* subspecies. 

### 3.2. Genome-Wide Distribution of SNPs

The 202 accessions were genotyped using next-generation whole-genome resequencing at a depth of 10×. SNPs were identified by comparing the genome sequence of the cultivated peanut Tifrunner (http://www.peanutbase.org, accessed on 20 October 2022). A total of 3,034,414 polymorphic SNP markers meeting the quality control criteria (MAF > 0.05 and integrity > 0.1) were identified using GATK, SAMtools, and ReseqTools. These SNP markers were employed for population structure definition, phylogenetic tree construction, and GWAS analysis. The distribution of SNPs was non-uniform across the 20 chromosomes, with the number of SNPs ranging from 56,837 in A08 to 209,586 in B09 (Figure 2A,B). 

### 3.3. Population Structure and Phylogenetic Tree

Principal component analysis of the 202 accessions grouped the panel into two groups, namely G1 and G2 (Figure 3A). The population structure was determined using PLINK and fastStructure software based on the genotypic data of the panel. When K = 2, the accession panel was divided into two subpopulations, G1 and G2 (Figure 3B), which corresponded to the two subspecies, *fastigiata* and *hypogaea*, respectively (Appendix A). Of the 202 accessions, 104 were classified into G1 and 98 were classified into G2. Most of the accessions (99 out of 104) in G1 belonged to subsp. *fastigiata*, while most of the accessions (84 out of 96) in G2 belonged to subsp. *hypogaea*. In subpopulation G1, 34.62% of the accessions was from southern China, 29.81% was from the Yangtze River region, 14.42% was from northern China, and 21.15% was from other regions. In subpopulation G2, 42.86% of the accessions originated from northern China, 13.27% originated from southern China, 27.55% originated from the Yangtze River region, and 16.33% originated from the other regions. The phylogenetic tree was constructed using the unweighted pair group method with arithmetic average (UPGMA) and maximum likelihood methods, showing that the 202 accessions were primarily divided into two clusters (Figure 3C). A total of 98 accessions were classified in cluster 1, which was similar to subpopulation G1, while 104 accessions were classified into cluster 2, which was similar to subpopulation G2. The results of the population structure and the phylogenetic tree were consistent. 

### 3.4. Identification of Loci Associated with Stem Rot Resistance Using GWAS

The Manhattan plot and quantile–quantile (Q-Q) plot revealed that 121 SNP markers were detected by the FarmCPU model (Figure 4), 104 SNP markers were detected by the GLM model (Appendix A), and 2 SNP markers were detected by the MLM model (Appendix A) based on the significance threshold of −log10 (1/3,034,414) = 6.48. The SNP markers detected by MLM were also detected by FarmCPU, while the SNP markers detected by GLM were also detected by FarmCPU. Therefore, SNP markers detected by FarmCPU were considered significant and used for further analysis. The 121 SNP markers explained 12.23% to 15.51% of the phenotype. The SNP markers were detected on 11 chromosomes, namely A03, A04, A07, and A10 in the A genome and B02, B03, B04, B05, B08, B09, and B10 located in the B genome (Figure 5). Of the 121 SNPs, 4, 116, and 1 SNP markers were identified in environment 2019-A, 2021-A, and 2021-B, respectively. 

Three significant SNPs on chromosome A04 were located in the genomic region from 120,110,031 bp to 122,646,296 bp, spanning 2.54 Mb, with the most significant SNP explaining 13.82% of the phenotypic variation. A total of 20 SNPs located on chromosome A10, from 17,040,050 bp to 66,057,341 bp, spanning approximately 49.02 Mb, with the most significant SNP having a phenotypic variation explained (PVE) of 14.57%. On chromosome B04, there were three significant SNPs in a genomic region from 137,391,933 bp to 139,559,492 bp, spanning 2.17 Mb, with the most significant SNP explaining 13.23% of the phenotypic variation. Thirteen significant SNPs were located on chromosome B05, in the genomic region from 15,951,630 bp to 155,781,790 bp, spanning 139.83 Mb, with the most significant SNP explaining 14.61% of the phenotypic variation. A set of 76 significant SNPs located on chromosome B08 in the genomic region from 20,423,727 bp to 106,147,465, spanning approximately 85.72 Mb, with the most significant SNP having a PVE of 15.51%. There was only one significant SNP distributed on chromosomes A03, A07, B02, B03, B09 and B10, respectively (Figure 5).

### 3.5. Identification of Putative Candidate Genes for Stem Rot Resistance 

The putative candidate genes surrounding the significant SNPs in the peanut genome sequence ±100 kb upstream and downstream of the significant SNPs were identified. A total of 292 genes were found surrounding 62 significant SNPs, while 27 genes were considered putative candidate genes associated with peanut stem rot resistance surrounding 23 significant SNPs. Some SNPs were associated with one candidate gene (Appendix A), such as B04-137391933 and B04-137407408, which were associated with *arahy. WVAI17*; B08-32412160, B08-3241216, and B08-32426809 were associated with *arahy. CE0NJE*; B08-32618254, B08-32668028, and B08-32668886 were associated with *arahy. 7DSU2J*; and B08-32830034, B08-32850779, B08-32883804, and B08-32904050 were associated with the *arahy. LUID63* gene. The potential candidate genes were located on five chromosomes, namely A04, B02, B04, B05, and B08. Functional annotation of the 27 potential candidate genes revealed that 12 genes were related to recognition, such as disease-resistance protein (TIR-NBS-LRR, LRR receptor-like kinase, protein kinase superfamily protein, receptor kinase, receptor-like protein, receptor-like protein kinase, receptor-like serine/threonine kinase). Thirteen genes were related to signal transduction, including ATPase, ATP-binding protein/serine and threonine kinase, Calcium-binding EF-hand family protein, glutathione S transferase, myb transcription factor, signal peptide peptidase, thioredoxin superfamily protein, transcription factor, WRKY family transcription factor, and zinc finger. Two genes were associated with defense response, namely peroxidase and terpene synthase (Appendix A and Table 4).

## 4. Discussion

Stem rot, caused by *A. rolfsii*, is a highly destructive and economically significant soil-borne fungal disease that affects the yield and quality of peanuts worldwide. Utilizing resistant varieties is the most cost-effective and efficient approach for managing this disease. In this study, we evaluated a panel of 202 peanut accessions, 81.8% of which originated from China, for their resistance to stem rot. The genotypic and phenotypic data of the panel were used for GWAS to gain insight into the genetic base of resistance.

Previous studies have evaluated resistance to stem rot in peanut germplasm. Mehan et al. (1995) [43] evaluated a total of 859 germplasm accessions, breeding lines, and interspecific hybrid derivatives for stem rot resistance and found 16 accessions were resistant. Bennett et al. (2020) [44] evaluated 71 accessions of the U.S. germplasm mini-core collection for stem rot resistance and found that 4 accessions had low levels of disease. Although some resistant accessions were identified in those reports, the difference in resistance between the two subspecies of peanut have not been compared. In this study, for the first time, resistance to stem rot in two subspecies was compared. A total of 110 accessions of subsp. *fastigiata* and 92 accessions of subsp. *hypogaea* were tested in three environments. The mean disease index values (47.86, 52.06, 58.47) of accessions in subsp. *fastigiata* was slightly lower than that (49.34, 53.31, 61.95) of accessions in subsp. *hypogaea* in 2019-A, 2021-A, and 2021-B, but the difference was not significant (Table 2). Resistant materials against stem rot were identified in both subsp. *fastigiata* and subsp. *hypogaea* (Appendix A); it suggested that resistance sources exist in both subspecies. Resistance to diseases in both subspecies have also been found in other reports. Ding et al. (2022) [26] found accessions with seed infection of *Aspergillus flavus*, and aflatoxin production resistance existed in both subsp. *fastigiata* and *hypogaea*. Yu et al. (2020) [27] also found accessions with aflatoxin production resistance in both subsp. *fastigiata* and *hypogaea*. The results implied that breeding for stem rot resistance could use sources from either subsp. *fastigiata* or subsp. *hypogaea*.

Resistance to stem rot in peanut varies among accessions originating from different geographical regions. In the current study, accessions from southern China exhibited the lowest disease index (48.89) compared to those from the Yangtze River region (56.28) and from northern China (51.98) across three environments (Table 3). It suggested that accessions from southern China are more resistant to stem rot compared to those from the other two regions. Among the 202 peanut accessions tested in this study, 7 were identified as resistant to stem rot, with 5 of them originating from southern China. In a previous study, Jiang et al. (2002) [45] evaluated 700 accessions for resistance to *A. flavus* invasion and found that all genotypes with a low invasion percentage were from southern China. These findings suggest that accessions from southern China may harbor more sources of resistance to some peanut diseases. 

Resistance to stem rot in peanut is a complex quantitative trait that is controlled by multiple genes. GWAS presents a powerful genetic mapping tool for the dissection of complex traits, including disease resistance, in many other crops [46,47,48,49], as well as in peanut. This study is the first report of the identification of SNPs associated with stem rot resistance using GWAS. A total of 121 SNP markers have been identified as being associated with stem rot resistance. Several earlier studies identified QTLs related to stem rot in peanut, all of which were detected by linkage analysis with RIL populations from a narrow genetic background. Regarding the different populations and methods in previous studies, those QTLs were found to be located in different genomes. Some studies detected QTLs in the A genome [10,13], while others found QTLs in the B genome [11]. In the present study, QTLs were identified in both the A and B genomes, which is consistent with the results of Luo et al. [12], who also detected QTLs in both the A and B genomes (Appendix A). This suggests that both genomes are a source of resistance to stem rot. Previous studies detected QTLs related to stem rot resistance on different chromosomes. Bera et al. (2016) [10] identified one QTL on chromosome A01. Dodia et al. (2019) [11] detected seven QTLs on chromosome B03, B04, B06, B08, and B10. Cui et al. (2020) [13] identified two QTLs on chromosome A01 and A05. Luo et al. (2020) [12] detected 33 QTLs on chromosome A01, A03, A04, A05, A06, A07, A08, A10, B03, B04, B05, B06, and B08. In the present study, significant SNPs related to resistance to stem rot have been identified on nine chromosomes reported in earlier studies, i.e., A03, A04, A07, A10, B03, B04, B05, B08, and B10, and two chromosomes (B02 and B09), previously unreported, have been identified in the present study (Appendix A). The genetic alleles related to stem rot resistance are located on different chromosomes, indicating that markers pyramiding might be an effective way to increase stem rot resistance in peanut breeding.

In this study, 121 significant SNPs were aligned to the reference genome of cultivated peanut Tifrunner with the previously detected QTLs linked to stem rot resistance [10,11,12,13]. The results show that 56 significant SNPs identified in this study were located in the QTL regions reported by others (Appendix A). For instance, SNP 04-121157345 and 04-122646296 detected in this study were located in the QTL qSR.A04-2 region identified by Luo et al. (2020) [12]. In addition, a total of 53 significant SNPs were detected in chromosome B08 (54475586-106147465) within the QTL q9DAI_S1B08.1 region [11], and another significant SNP marker B10-80733496 identified in this study was located within the region of QTL q9DAI_S3B10.2 [11]. The results suggest these significant SNPs identified in this study are reliable markers associated with stem rot resistance in peanut. Among these markers, 04-121157345 was located in the QTL qSR.A04-2 region identified in environment 2021-A, while another significant SNP marker 04-122646296 was identified in environment 2021-B. The findings suggest that the QTL qSR.A04-2, detected in two environments, is considered a conserved QTL. These reliable SNPs related to stem rot resistance in peanut can help establish diagnostic markers for breeding programs and aid in the discovery of putative candidate genes associated with stem rot resistance in peanut.

## 5. Conclusions

In summary, our study found no significant difference in peanut stem rot resistance between subsp. *fastigiata* and subsp. *hypogata*. However, accessions from southern China exhibited higher resistance compared to those from other regions. We identified a total of 121 significant SNPs associated with stem rot resistance in peanut using GWAS and annotated 27 candidate genes that may confer resistance to peanut stem rot.

## Figures and Tables

**Figure 1 genes-14-01447-f001:**
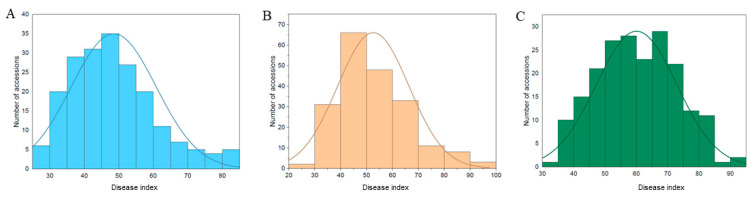
Frequency distribution of stem rot resistance of peanut in 2019-A, 2021-A, and 2021-B. (**A**): the experiments conducted at Wuchang Experimental Station in 2019; (**B**): the experiment carried out at Wuchang Experimental Station in 2021; (**C**) the experiment conducted at Yangluo Experimental Station in 2021.

**Figure 2 genes-14-01447-f002:**
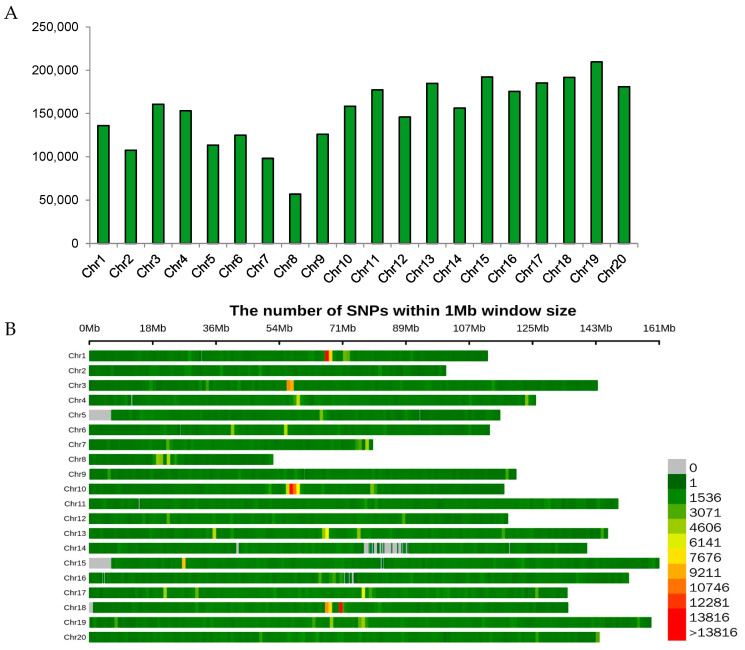
Single-nucleotide polymorphisms (SNP) distribution across the 20 chromosomes of the cultivated peanut. (**A**) The number of SNPs across each chromosome in peanut genome. (**B**) SNP density on each chromosomal pseudomolecule of peanut. The horizontal axis shows the length of the SNPs per 1 Mb window.

**Figure 3 genes-14-01447-f003:**
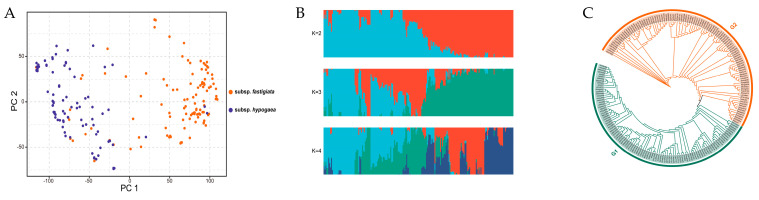
Principal analysis, structure population, and phylogenetic tree of the 202 accessions. (**A**) Principal analysis of peanut panel and divided into two groups. (**B**) Structure population results at K = 2, 3, and 4 of the 202 peanut accessions. (**C**) Phylogenetic tree constructed by UPGMA and divided into two clusters, G1 (blue line) and G2 (brown line).

**Figure 4 genes-14-01447-f004:**
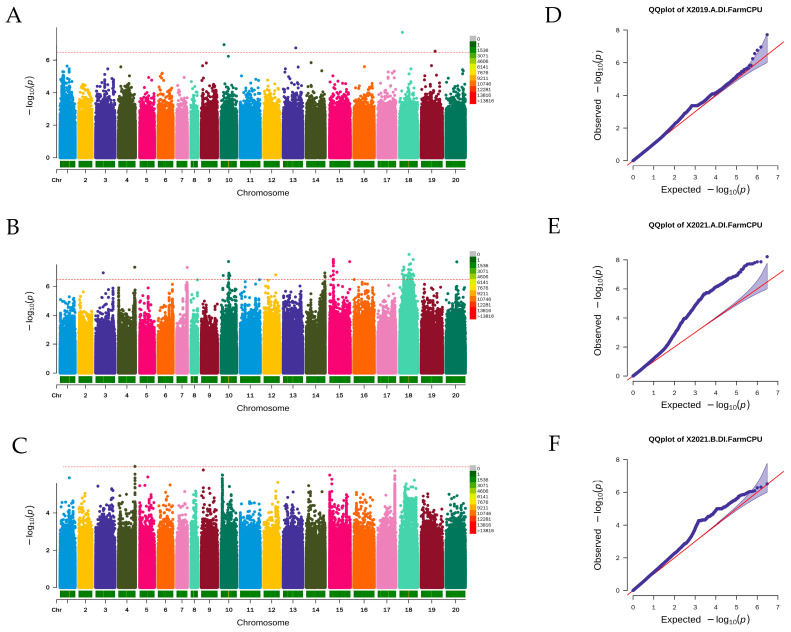
Manhattan and Q-Q plots for stem rot resistance in peanut in 2019-A, 2021-A, and 2021-B using FarmCPU model. The horizontal line indicates the following genome-wide significant threshold: −log10 (1/3,034,41) = 6.48. (**A**) Manhattan plot of 2019-A. (**B**) Manhattan plot of 2021-A. (**C**) Manhattan plot of 2021-B. (**D**) Q-Q plot of 2019-A. (**E**) Q-Q plot of 2021-A. (**F**) Q-Q plot of 2021-B.

**Figure 5 genes-14-01447-f005:**
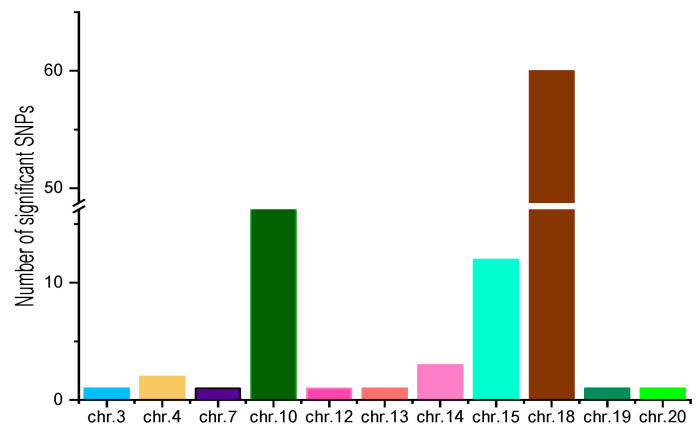
The significant SNPs conferring stem rot resistance in peanut identified using the FarmCPU model.

**Table 1 genes-14-01447-t001:** Phenotypic variation of disease index of 202 peanut accessions to stem rot across three environments.

Environment	Minimum	Maximum	Mean	SD	CV	Skew	Kurt	*H*^2^ (%)
2019-A	27.03	83.75	48.52	12.33	0.25	0.72	0.23	67.1
2021-A	28.71	97.34	52.62	13.77	0.26	0.85	0.50	
2021-B	34.26	94.29	60.06	12.85	0.21	0.21	−0.65	

Note: 2019-A means the data collected in 2019 at the Wuchang Experimental Station; 2021-A means data collected in 2021 at the Wuchang Experimental Station; 2021-B means data collected in 2021 at the Yangluo Experimental Station.

**Table 2 genes-14-01447-t002:** Disease index of two subspecies in 202 peanut accessions.

Subspecies	Environment			
	2019-A	2021-A	2021-B	Mean
subsp. *fastigiata*	47.86 ± 12.23 a	52.06 ± 13.52 a	58.47 ± 12.80 a	52.8 ± 9.59 a
subsp. *hypogaea*	49.34 ± 12.47 a	53.31 ± 14.11 a	61.95 ± 12.71 a	54.89 ± 9.89 a

Note: 2019-A means the data collected in 2019 at the Wuchang Experimental Station; 2021-A means data collected in 2021 at the Wuchang Experimental Station; 2021-B means data collected in 2021 at the Yangluo Experimental Station. Means followed by the same letter within a column are not significant different (*p* ≤ 0.05) according to Dunnett’s Test.

**Table 3 genes-14-01447-t003:** Disease index variation of accessions originated from different areas.

Originated Place	Environment			
	2019-A	2021-A	2021-B	Mean
Southern China	45.39 ± 10.60 a	46.20 ± 11.22 a	54.92 ± 11.54 a	48.89 ± 8.22 a
Yangtze River region	51.17 ± 14.48 b	54.59 ± 15.03 b	63.08 ± 14.75 b	56.28 ± 11.41 b
Northern China	45.68 ± 9.29 a	50.09 ± 9.47 ab	60.04 ± 10.42 b	51.98 ± 5.65 a
Other countries	52.64 ± 13.06 b	61.71 ± 15.10 c	62.09 ± 13.07 b	58.82 ± 10.40 c

Note: 2019-A means data collected in 2019 at the Wuchang Experimental Station; 2021-A means data collected in 2021 at the Wuchang Experimental Station; 2021-B means data collected in 2019 at the Yangluo Experimental Station. Means followed by the same letter within a column are not significant different (*p* ≤ 0.05), while followed by different letters within a column are significant different (*p* ≤ 0.05) according to Dunnett’s Test.

**Table 4 genes-14-01447-t004:** Putative candidate genes associated with stem rot resistance in peanut.

Function	Annotation	Number of Genes
Recognition (12)	disease-resistance protein (TIR-NBS-LRR)	1
	LRR receptor-like kinase	1
	protein kinase superfamily protein	2
	receptor kinase	2
	receptor-like protein	1
	receptor-like protein kinase	3
	receptor-like serine/threonine kinase	2
Signal transduction (13)	ATPase	1
	ATP-binding/protein serine/threonine kinase	1
	calcium-binding EF-hand family protein	1
	glutathione S-transferase	1
	myb transcription factor	1
	signal peptide peptidase	1
	thioredoxin superfamily protein	1
	transcription factor	1
	WRKY family transcription factor	2
	zinc finger family protein	3
Defense (2)	peroxidase superfamily protein	1
	terpene synthase	1

## Data Availability

Not applicable.

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
