# Peer review of "Dissection of the Genetic Basis of Resistance to Stem Rot in Cultivated Peanuts (Arachis hypogaea L.) through Genome-Wide Association Study"

_genes, 2023, doi:10.3390/genes14071447_

Round 1

Reviewer 1 Report

In this study ,a whole-genome resequencing approach was used to mine high-quality SNPs distributed throughout 202 peanut accessions. Genomic regions associated with stem rot disease were identified through GWAS, and potential candidate genes within the QTLs were identified. This study discovered SNP markers and candidate genes associated with stem rot resistance. In my opinion it is a very significant contribution to the field of agriculture. However, there are some concerns I would like to address:

1. It is not clear to me why the authors used oat grains as a source for inoculation with stem rot. pls. add a detailed description of this issue, supported by the appropriate citation.

2. For GWA analysis, authors performed 3 models, namely, the generalized linear model (GLM), the mixed linear model (MLM), and fixed and random model circulating probability unification (FarmCPU).  I think a brief description of these models should be added to clarify their significance in the analysis. 

3. I noticed that both tables and figures were not placed in their appropriate  position which coincide with their citation in the text. pls. check this point.

4. is susceptible indicator variety ( Zhonghua 9 ) used as a control? please clarify the reasons and cite them.

5. References were not listed by numbers, pls. check

writing style and mechanics in grammar should be double -checked by a professional editing service

Author Response

Dear reviewer,

Thank you very much for your precious comments to the manuscript, I have made the changes according to your suggestion point to point.

  1. It is not clear to me why the authors used oat grains as a source for inoculation with stem rot. pls. add a detailed description of this issue, supported by the appropriate citation.

Response: Since Athelia rolfsii do not produce spores in its life cycle, it is difficult to inoculate plants with mycelium discs or sclerotia. Some scientists found that it is a convenient way to inoculate plants with mycelium of A. rolfsii infested oat grains, so we also use the inoculum in the present study. I added the appropriate citations in the manuscript, for instance, Fan et al. 2020 and Shokes et al. 1996.

  1. For GWA analysis, authors performed 3 models, namely, the generalized linear model (GLM), the mixed linear model (MLM), and fixed and random model circulating probability unification (FarmCPU).  I think a brief description of these models should be added to clarify their significance in the analysis. 

Response: I have added a brief description of these models in the manuscript.

  1. I noticed that both tables and figures were not placed in their appropriate position which coincide with their citation in the text. pls. check this point.

Response: Thank for your opinion. I have already put the tables and figures in the right citation in the text.

  1. is susceptible indicator variety ( Zhonghua 9 ) used as a control? please clarify the reasons and cite them.

Response: Yes, I just used Zhonghua 9 as an indicator. Now, I have already removed this part from the manuscript.

  1. References were not listed by numbers, pls. check

Response: Thank you for your opinion. I have listed the references by numbers.

Reviewer 2 Report

Dear Authors

All references used in the manuscript are numeric numbers while the reference section at the end of the manuscript does not have them. It's very hard to keep track of references when I need to check some information on the original reference.   

Page2: in the Plant materials and field evaluation of stem rot resistance section, from where the plant materials was obtained? Please mension the seed supplier.

Page 3: (300 mg autoclaved oat grains in a 1 L flask) is 300g or 300 mg?

Author Response

Dear reviewer,

Thank you very much for your precious comments to the manuscript, I have made the changes according to your suggestion point to point.

  1. All references used in the manuscript are numeric numbers while the reference section at the end of the manuscript does not have them. It's very hard to keep track of references when I need to check some information on the original reference.   

Response: Thank you for your opinion. I have listed the references by numbers.

Page2: in the Plant materials and field evaluation of stem rot resistance section, from where the plant materials were obtained? Please mention the seed supplier.

Response: Those plant materials were collected by our group from China and other countries. We multiplied the seeds and used in the field evaluation.

Page 3: (300 mg autoclaved oat grains in a 1 L flask) is 300g or 300 mg?

Response: Yes, it should be 300 g, I have made the modification.

Reviewer 3 Report

1. The evaluated paper is valuable in scientific terms. Its biggest drawback, however, is the lack of adaptation to the requirements of the Genes journal.

2. Acronyms used in tables should be explained below the tables, e.g. 2019-A.

3. Tables 1 to 4 should be placed in the Results section after their first citation in the text of the manuscript. For example, Table 1 should be placed after Figure 1.

4. Figures (especially those consisting of several smaller figures) are difficult to read.

5. Figure 3 - individual figures should be marked: A, B and C.

6. The list of literature in the References section should be numbered so that the reader can easily find the source of literature cited by the authors.

7 No Figure S1 title.

8 In "Table S2. Pearson correlation analysis ...." - the authors did not provide a correlation coefficient between 2021-B and 2021-A.

Author Response

Dear reviewers,

Thank you very much for your precious comments to the manuscript, I have made the changes according to your suggestion point to point.

  1. The evaluated paper is valuable in scientific terms. Its biggest drawback, however, is the lack of adaptation to the requirements of the Genes journal.

Response: I have use the genes template to modify the manuscript.

  1. Acronyms used in tables should be explained below the tables, e.g. 2019-A.

Response: I have added the notes below the tables, and explain 2019-A and etc.

  1. Tables 1 to 4 should be placed in the Results section after their first citation in the text of the manuscript. For example, Table 1 should be placed after Figure 1.

Response: I have made the change, put the figures and tables just after first citation.

  1. Figures (especially those consisting of several smaller figures) are difficult to read.

Response:I have added the letters in each smaller figures to make it is easier to read.

  1. Figure 3 - individual figures should be marked: A, B and C.

Response:I have added the A, B, C to the individual figures in Figure 3.

  1. The list of literature in the References section should be numbered so that the reader can easily find the source of literature cited by the authors.

Response: Thank you for your suggestion, I have numbered the references.

  1. No Figure S1 title.

Response: The figure S1 title was at the last of the supplementary list. I have changed the position of figure S1 title according to the genes template.

  1. In "Table S2. Pearson correlation analysis ...." - the authors did not provide a correlation coefficient between 2021-B and 2021-A.

Response: I have added the correlation coefficient between 2021-B and 2021-A in the text.

Round 2

Reviewer 2 Report

Dear Authors

The manuscript finds a new promising 27 candidate genes confer resistance to stem rot disease caused by Athelia rolfsii. All these genes need to be validated after cloning them in the near future.

The paper looks in good shape.